# Improvement of Nutritional Quality of Tomato Fruit with *Funneliformis mosseae* Inoculation under Greenhouse Conditions

**Fazal Ullah [1], Habib Ullah [2] , Muhammad Ishfaq [3] , Syeda Leeda Gul [4], Tanweer Kumar [5] and Zhifang Li [1,*]**

1   Beijing Key Laboratory of Growth and Developmental Regulation for Protected Vegetable Crops, Department of Vegetable Science, College of Horticulture, China Agricultural University, Beijing 100193, China
2   Department of Chemistry, Government Post Graduate College, Abdul Wali Khan University, Mardan 23200, KPK, Pakistan
3   College of Resources and Environmental Sciences, National Academy of Agriculture Green Development, Key Laboratory of Plant-Soil Interactions, Ministry of Education, China Agricultural University, Beijing 100193, China
4   China State Key Laboratory of Plant Physiology and Biochemistry, College of Biological Sciences, China Agricultural University, Beijing 100193, China
5   Sugar Crops Research Institute, Agriculture livestock Fisheries and Cooperative Department, Mardan 23200, KPK, Pakistan
*   Correspondence: zhifangli7@cau.edu.cn

**Abstract:** Long-term soil mining with extensive cultivation practices and traditional breeding methods have declined the flavor and nutritional value of tomatoes. Apart from important mineral nutrients (i.e., nitrogen, phosphorus, and potassium), fungi known as arbuscular mycorrhizae (AM) can considerably improve the quality of agricultural production through higher phosphate uptake. Using hydroponically cultured commercially available tomato cultivars, we investigated the possible effects of mycorrhizae in improving the nutritional quality of tomato fruit. *Funneliformis mosseae* (syn. *Glomus mosseae*)-inoculated tomato plants were grown on a 1:1 mixture of peat and vermiculite, and different phosphorus levels were applied. RNAseq and metabolites were studied to confirm the relative gene expression and metabolites in fruit tissues. The results showed that AM inoculation with low phosphorus can significantly improve important fruit-quality traits such as free amino acids, lycopene (47.9%), and β-carotene (29.6%) without compromising the yield. Further, differentially expressed genes (DEGs) were identified by comparing the nutritional and ripening potential of fruits produced by mycorrhizal and non-mycorrhizal plants. Notably, carotenoids and sugars (BRIX values) were found to be higher in mycorrhized plants in contrast to non-mycorrhized plants. Therefore, the current study suggests mycorrhization as a promising approach for the production of high-quality tomato fruit for human consumption.

**Keywords:** phosphate fertilizer; yield; free amino acids; fruit quality; BRIX value; carotenoids; arbuscular mycorrhiza

## 1. Introduction

Tomato (*Solanum lycopersicum*) is one of the most important vegetables for human consumption, especially for the so-called Mediterranean diet, which is low in fat. With an annual production of 177 million metric tons and individual consumption of 18 kg, it is the most economically significant fleshy fruit vegetable in the world [1]. Tomatoes are used in a wide range of food and beverage products, not only in their fresh form but also in preserves and juice concentrates. Though the plants can be commercially grown in greenhouses and open fields, greenhouses are considered ideal for tomato production due to more consistent harvesting throughout the year. However, the fruit flavor and nutritional value have declined in the past decades as a direct result of conventional breeding [2]. In addition, large quantities of mineral fertilizers are required to gain high tomato yields [3,4].

Rock phosphate, however, is a limited resource that is expected to largely decline in the next 40 to 70 years [5]. Thus, it is necessary to develop strategies to improve the fertilizer-use efficiency of crop plants to meet the growing demand for more food with a high content of nutrients [3,4,6,7]. The beneficial effects of arbuscular mycorrhizal fungi (AMF) and other root symbionts on the overall performance of crop plants have gained attention in the last few decades. Arbuscular mycorrhiza (AM) is a type of mutualistic symbiosis relationship between a plant and a fungus member of the phylum Glomeromycotina (formerly Glomeromycotina [8]). This symbiotic relationship plays an important role in the evolution of land plants, most likely because of AMF's enhanced phosphate uptake from the rhizosphere [9]. The fungus can colonize the root cortexes, grow intercellular hyphae, and produce arbuscules as a result of an intricate molecular interaction with its host plant. Arbuscules are highly branched structures that are created within the cortex cells [10]. This process takes place as a result of the interaction between the fungus and its host plant. The fungus, through its expanded mycelium, makes it easier for the plant to extract nutrients from the surrounding soil, particularly nitrogen and phosphorus [11,12]. The mycorrhizal pathway can deliver the plant-required phosphate by reducing the need for the direct root pathway [13]. However, the nutrient-uptake efficiency varies across different plant species. The beneficial symbiotic fungus receives carbohydrates from the plant in exchange, which are then metabolized by the fungus into hexoses and fatty acids. [14]. This interaction not only enhances the nutrient availability for plants, but also increases the plant's resistance to pathogens, water/salt-stress, and pollutants [15–17]. Other beneficial effects of AM that have been studied extensively include an acceleration at the beginning of flowering as well as an increase in both the yield and the number of fruits that are fit for human consumption [18–22]. Thus, AMF are increasingly being incorporated into plant production systems due to a lower use of chemical fertilizers and pesticides, which enhances their contribution to environmentally friendly food production [23]. Therefore, microbial inoculants such as AMF as "biostimulants" might be a promising solution for the provision of nutritious food for human consumption [24]. Mycorrhization promotes earlier flowering and maturation of the fruit by optimum nutrient supply during plant growth [25]. However, in some of the studies root colonization in tomato plants did not increase the plants' overall fruit yield [18,26]. Furthermore, there is evidence that interacting with AMF improves the quality of nutrients for tomato plants by increasing amount of citric acid, carotenoids, and amino acids, as well as the fruit's ability to fight free radicals [18–28]. In addition, tomatoes derived from mycorrhized plants have greater anti-estrogenic properties, do not exhibit genotoxic effects [29], and do not present an increased allergenic risk to human beings [30]. This result, however, was not consistent regardless of the tomato cultivar or fungus species used in the previous studies. In addition, the majority of the research was carried out in laboratory conditions, which are very different from tomato production for commercial purposes. A large number of greenhouse tomato growers prefer "soilless" production because it produces high yields and maintains a high fruit quality. These greenhouse cultivation systems enable greater control over the amount of fertilization and irrigation that is applied. On the other hand, it is unknown if AMF can improve the tomato fruit quality that is grown in these conditions. As a consequence of this, the purpose of our research was to determine the ideal conditions for the growth of tomatoes and the production of AM in greenhouses. We combined commercial tomato cultivars (cv. Rio Grande and Nadir) with the model AM fungus species *F. mosseae*, which is capable of adapting to foreign plant species and is widely distributed across the world [31]. Further, its genome has been sequenced [32], and cultivating and maintaining *F. mosseae* in trap cultures is very easy. To find markers related to tomato quality, it is also necessary to identify genes that are differentially expressed between fruits from non-mycorrhized plants and fruits from mycorrhized plants. Our study provided a scientific basis for fruit metabolites and important quality traits to better understand the possible potential of mycorrhizae in improving the nutritional quality of tomatoes grown under commercial greenhouse conditions.

## 2. Materials and Methods

### 2.1. Plant Material and Growth Conditions

Commercially available tomatoes cv. Rio Grande and cv. Nadir were cultivated in the Kalar Kahar commercial greenhouse, Pakistan. The experiment was started from May 2021 to August 2021 and from February 2022 to April 2022 in $22 \times 13$ cm pots filled with a 1:1 mixture of peat and vermiculite. A 10 mL volume of nutrient solution containing 2.5, 6.5, or 10.5 mM phosphate [33] were applied to the plants two times per week. One-week old seedlings of cv. Rio Grande and cv. Nadir were inoculated with *F. mosseae* strain BEG12 [34]. Mycorrhiza inoculum, a commercial powder containing highly concentrated mycorrhiza propagules (90.000/g), was used for inoculation at a rate of $1 \text{ g L}^{-1}$ substrate. Continuous fertilization was achieved through the use of a drip irrigation system. A basic dose of macro- and micronutrients was applied as follows: 2 mM Ca $(NO_3)_2.4H_2O$, 2.5 mM CO $(NH_2)_2$, 5 mM $KNO_3$, 1 mM $KH_2PO_4$, 1 mM $MgSO_4.7H_2O$, 23 μM $H_3BO_3$, 10 μM $MnSO_4.H_2O$, 4.5 μM $ZnSO_4.7H_2O$, 0.8 μM $CuSO_4.5H_2O$, 0.5 μM $H_8MoN_2O_4$, and 16 μM EDTA-FeNa.$3H_2O$. The nutrient solution was provided during the first few months of plant growth, with as few as three irrigation cycles and as many as thirty after that. Since the greenhouse substrate should demonstrate an overall drainage of 30%, the number of irrigation cycles was regulated by the extent of drainage. The drain's electric conductivity was measured to ensure the plants received the right amount of water and fertilizer during the hot summer months. Throughout the maturation of the tomatoes, three bumblebee boxes were set free to ensure reliable pollination. The plants were hung on hooks for optimal growth, and the adventitious shoots were removed twice weekly. The fruits were allowed to ripen by selectively pruning leaves at a rate of no more than three leaves per week, and each truss was matured separately. The tomatoes were picked from their panicles as soon as they were ripe, and the two or three leaves that had grown above them were cut off. The plants were hung further along the ceiling as their main stems grew, eventually reaching a length of five meters, to ensure the continuous emergence of flowers and fruits.

### 2.2. Transcriptomics and Quantitative Real-Time PCR (qPCR)

During harvesting, a total of eight Rio Grande/Nadir plants from both mycorrhizal and non-mycorrhizal varieties and fruits were flash-frozen in liquid nitrogen and stored. To combine two fruits into a single sample, the pericarps were lyophilized, homogenized, and pooled to create 16 samples. The isolation of RNA was carried out using the RNeasy Plant Mini Kit, which required only a single elution in 30 L water. DNA was extracted from the fruit tissues using a DNA-free TM Kit following a standard DNase treatment and a rigorous DNase treatment. The latter treatment included a 1:1 RNA dilution and 2 L DNase. Using an Agilent Bioanalyzer 2100, we found RIN factors greater than 8. Subsequently, RNAseq was performed with at least 1 g of RNA by Novogene UK Company Limited, and 150 bp paired-end reads were recorded. The bioinformatics for the DEG analysis was carried out using Novogene and the DESeq2 R package, with read counts as input [35] and the Benjamini–Hochberg method for *p*-value adjustment. The DEGs' log10 (FPKM + 1) values were used to perform a hierarchical clustering analysis across all comparison groups. The ArrayExpress database, accession number E-MTAB-9419, contains the sequence data in the form of FASTQ files from the RNAseq experiment.

To verify the RNAseq data, cDNA was produced from 1 g of RNA extracted from fruit tissues using oligo (dT) primers and the ProtoScript II First Strand cDNA Synthesis Kit. The quantitative polymerase chain reaction (qPCR) was carried out by diluting cDNA to 1:20 using EvaGreen QPCR Mix II without ROX along with gene-specific primers (Supplemental Table S1). To carry out qPCR using a CFX Connect cycler, the following procedure was followed: 15 min at 95 °C, followed by 40 cycles of 15 s at 95 °C and 30 s at 56 °C, and finally, a melt curve was generated using fluorescence detection between 60 degrees Celsius and 95 degrees Celsius for 1 s at every 0.5-degree increment. Using the Cq method [36], the expression levels of all of the genes of interest were compared to those of TIP41.

### 2.3. Determination of BRIX Values

Firstly, 1 mL of distilled water was added to the non-pooled lyophilized material of red fruits derived from both mycorrhized and non-mycorrhized plants (cv. Rio Grande/Nadir). Then, to dissolve the soluble solids, the samples were first thoroughly mixed, then incubated at room temperature for 3 h, and finally centrifuged for one minute at 7000 g. A refractometer was used to measure the BRIX content of the collected supernatant (juice).

### 2.4. Amino Acid, Carotenoid, and Mineral Determination

Freeze-dried samples were shaken at 70 °C for 30 min in 400 μL methanol. Subsequently, they were centrifuged (18,000 g, 10 min, 4 °C) after adding 200 μL $CHCl_3$ and 400 μL distilled water, and an 80 μL aliquot of the aqueous methanol phase was vacuum-dried and re-dissolved in 10 μL water. In order to derivatize the solute, an AccQ Fluor kit was used (Waters, Eschborn, Germany). Finally, amino acids were quantified using a high performance liquid chromatographic method coupled with an ultraviolet (HPLC-UV) detector (Waters UPLC with diode arraydetector; Eschborn, Germany) after serially diluting reference amino acid mixtures. Waters suggested AccQ-Tag Ultra. 1.7 μm, 2.1×100 mm UPLC column separated 1 μL aliquot. Empower discovered amino acids at 266 and 473 nm (Waters). The lower limit of detection (LOD) for 19 different amino acids was in the range of 0.01–0.05 μg mL$^{-1}$ whereas their limit of quantification (LOQ) was 0.03–0.15 μg mL$^{-1}$, as claimed by the instrument's manufacturer. The obtained coefficient of determination ($R^2$) for all amino acids was higher than 0.95, which indicates the goodness of fit among the variables.

Carotenoids were detected using HPLC-UV (Waters UPLC with diode array detector; Eschborn, Germany). Briefly, 600 μL 10 μM KOH-acetone was extracted 150 mg. The pellet was centrifuged and re-extracted twice with 500 μL extraction solvent after shaking for 10 min at RT and re-centrifuged. HPLC used 150 × 4.5 mm C30 columns and precolumns. As eluents, 80% methanol with 0.2% ammonia acetate and tert-butylmethylether were used (eluent C). After 1.0 mL min$^{-1}$ injection, 85% A, 5% B, and 10% C lasted 3 min, followed by 75% A, 5% B, and 20% C. Then, 15% A, 5% B, and 80% C followed 1 min later, followed by a 5 min re-equilibration. The LOD of HPLC-UV for β-carotene, lycopene, lutein, and zeaxanthin was 0.05, 0.01, 0.02, and 0.01 μg mL$^{-1}$, respectively. The LOQ of the instrument for β-carotene, lycopene, lutein, and zeaxanthin was 0.15, 0.03, 0.06, and 0.03 μg mL$^{-1}$, respectively. The obtained coefficient of determination ($R^2$) for β-carotene was 0.99.

Finely ground fruit tissues were weighed into PTFE digestion tubes with strong nitric acid (0.5 mL; 67–69%) for mineral analysis. The samples were digested in a high-pressure microwave reactor after 4 h of incubation (Ultraclave 4; MLS). De-ionized water diluted the digested contents to 8 mL in Greiner centrifuge tubes. Software 3.1.2.242 inductively coupled plasma–mass spectrometry (Sector Field High Resolution (HR)-ICP-MS, ELEMENT 2, Thermo Fisher Scientific, Dreieich, Germany) detected mineral elements, which were quantified as described earlier [4]. For different mineral elements (B, Mo, P, Ca, Mn, Fe, Ni, Cu, Zn, Na, Mg, S, and K), the LOD of ICP-MS was in the range of 0.002–0.009, while their LOQ was 0.003–0.030 μg L$^{-1}$. The coefficient of determination ($R^2$) for all mineral elements was calculated higher than 0.97.

### 2.5. Statistical Analyses

The analysis of variance (ANOVA) with a single factor was carried out to examine all data. Tukey's test was utilized for multiple comparisons. The significance level was set at a *p*-value of less than or equal to 0.05. All statistical tests and analyses were performed using SPSS 25.

## 3. Results

### 3.1. Effects of Mycorrhization on the Transcriptome of Tomato Fruit

To date, no significant effects of mycorrhizal inoculation on greenhouse tomatoes' total yield is reported. In our study, the total plant yield was the same for inoculated and non-inoculated plants (Figure S1).

The transcriptomes of both green and red fruits were examined to understand the possible physiological effects on red fruits. RNAseq showed that each sample had between 38.4 and 60.6 million mapped reads. Hierarchical clustering of the genes that were differentially expressed in the fruit tissues produced by mycorrhized and non-mycorrhized plants were further grouped to underline the effect of mycorrhization on the expression pattern of genes. The heatmap showed two distinct categories for the fruits, represented by the colors green and red, respectively. This demonstrates that there are substantial differences between the two stages in terms of the expression of genes (Figure 1). On the other hand, neither the green nor the red stages contained any sub-clusters that displayed the effects of mycorrhization. Based on the results, mycorrhization did not largely alter the way in which genes were expressed in fruits.

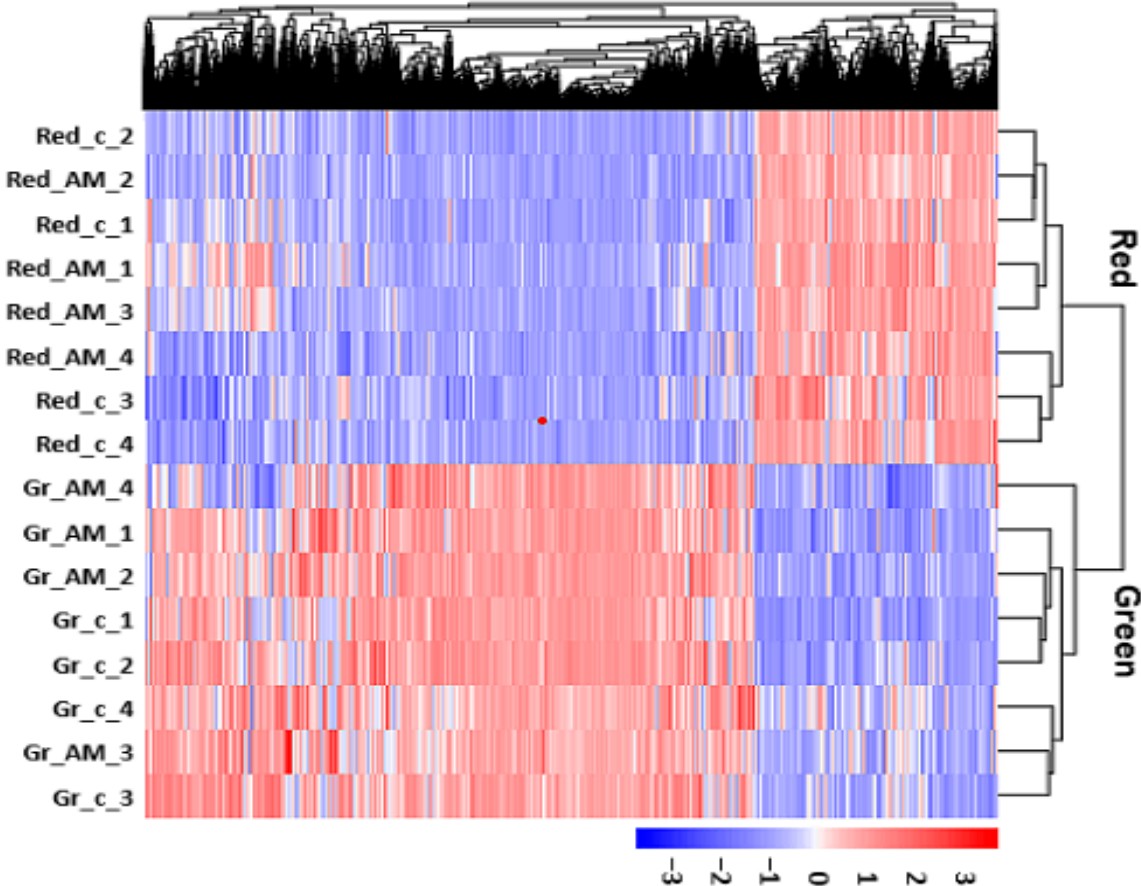

**Figure 1.** Heatmap displaying differentially expressed genes in hierarchical clusters (DEGs). Clustering was carried out utilizing the log10 (FPKM + 1) values of differentially expressed genes (DEGs) between green and red fruits of both mycorrhizal (AM) and non-mycorrhizal (c) plants (*n* = 4, $p \leq 0.05$). The Z score of the gene is column-scaled, and the colors represent range from low (blue) to high (red).

The number of genes with different levels of expression (also known as DEGs) in individual comparisons demonstrated that there were 13,004 DEGs between green and red fruits, but there were only 80 and 51 DEGs between green and red fruits from mycorrhized and non-mycorrhized plants, respectively. This indicated that mycorrhizal plants have

a greater genetic diversity than non-mycorrhizal plants (Supplemental Table S2). Each DEG found in red fruits exhibited fold expression changes that were lower than two. In addition, variations in gene expression during the red stage of development might have occurred too late to affect the red fruit's metabolome. Therefore, DEG levels were focused in green fruits. Mycorrhizal plants were found to contain all of these genes, except for two. To verify the findings from RNAseq, qPCR was performed on a total of 15 genes. The findings demonstrated that the selected genes were higher-expressed in the green fruits that originated from mycorrhizal plants (except ACC-oxidase5) than in the green fruits that belong to non-mycorrhizal plants (Table 1). However, the overall gene expression was considerably lower in non-mycorrhizal plants.

**Table 1.** Selected DEGs' expression levels in green fruits of plants with and without mycorrhizae.

| Gene Name | Solyc No | +AM | −AM | Log2FC | +AM | | −AM | | Log2FC |
|---|---|---|---|---|---|---|---|---|---|
| | | FPKM | FPKM | | rEx 2 | SE | rEx 2 | SE | |
| Aminotransferase | Solyc01g007940.3 | 2.818 | 0.050 | 5.7 | 0.0560 | 0.0390 | 0.0007 | 0.0008 | 6 |
| Dehydrin | Solyc02g084840.3 | 41.327 | 0.080 | 9.1 | 0.6888 | 0.6240 | 0.0020 | 0.0017 | 8.5 |
| Zinc finger TF 50 | Solyc07g053750.1 | 4.943 | 0.140 | 5.1 | 0.0550 | 0.0412 | 0.0020 | 0.0006 | 5 |
| PIN5 | Solyc01g068410.3 | 0.79 | 0.0522 | 4.2 | 0.0070 | 0.0056 | 0.0065 | 0.0003 | 4.3 |
| LEA | Solyc02g062770.2 | 11.522 | 0.14 | 6.5 | 0.2100 | 0.1943 | 0.0013 | 0.0014 | 7.2 |
| ACC-oxidase5 | Solyc07g026650.3 | 4.423 | 10.090 | −1.3 | 0.0100 | 0.0034 | 0.0266 | 0.0076 | −1.3 |
| Oleosin | Solyc06g069260.1 | 2.814 | 0.261 | 4.1 | 0.0230 | 0.0208 | 0.0008 | 0.0008 | 5 |
| ERF13 | Solyc04g080910.1 | 0.995 | 0.059 | 4.2 | 0.014 | 0.0070 | 0.0003 | 0.0003 | 5.8 |
| LEA 4 | Solyc10g078780.2 | 5.673 | 0.053 | 6.6 | 0.1120 | 0.1018 | 0.0009 | 0.0009 | 7.2 |
| 2S albumin seed storage | Solyc07g064210.2 | 16.756 | 0.147 | 6.9 | 0.1690 | 0.090 | 0.0012 | 0.0012 | 7.2 |
| Desiccation-related | Solyc05g053350.3 | 13.991 | 0.567 | 4.9 | 0.2051 | 0.1110 | 0.007 | 0.0023 | 5.1 |
| malic enzyme | Solyc12g008430.2 | 5.042 | 0.048 | 7.06 | 0.1346 | 0.1031 | 0.0369 | 0.0074 | 1.9 |
| MADS-box TF | Solyc04g078300.3 | 1.172 | 0.020 | 5.8 | 0.0141 | 0.0113 | 0 | 0 | ∞ |
| bZIP TF | Solyc10g080410.2 | 0.277 | 0.008 | 5.2 | 0.0111 | 0.0060 | 0.0002 | 0.0001 | ∞ |
| Vicilin | Solyc02g085590.3 | 6.88 | 0.28 | 5.3 | 0.048 | 0.1766 | 0.0060 | 0.0041 | 5.5 |

RNAseq analysis was used to determine which DEGs to choose. RNAseq data show FPKM values as means (*n* = 4) and log2 fold changes (FC); 2 validation using RT-qPCR analysis shows relative expression (rel. expr.) with SlTIP41, along with the standard error (SE). Student's *t*-test with $p < 0.05$ showed no significant differences in RT-qPCR results.

Under greenhouse conditions, the data indicated that arbuscular mycorrhization had only a moderate impact on the fruit transcriptome of tomato plants.

### 3.2. Effect of AMF on the Fruit Quality of Tomato

Although AM has a small effect on the tomato fruit transcriptome, the important compounds that might make red fruits healthier or taste better were differentially expressed. The BRIX value, which measures the amount of dissolved solids, mostly sucrose, is a simple way to understand fruit sweetness. The analysis showed that red fruits from mycorrhized plants contained higher BRIX values (Table 2). Thus, the data suggested that mycorrhization makes tomatoes sweeter. Further, mineral elements, β-carotenoids, and amino acid concentrations were measured and compared in green and red fruits grown with and without mycorrhizae (Table 2, Figure S2 and Figure 2). The data of the mineral analysis revealed no significant difference between the green and red fruits produced by plants with and without mycorrhizae (Figure S2). When comparing the two stages of development in terms of β-carotenoids, a significant difference was found in the concentrations of lutein, zeaxanthin, lycopene, and β-carotene. On the other hand, there were no noticeable differences found between the fruits produced by mycorrhizal and non-mycorrhizal plants when they were at the same stage of development. Despite this, the lycopene and β-carotene contents were significantly increased in red fruits as a result of mycorrhization (Table 2).

**Table 2.** Carotenoid contents of green and red fruits with and without mycorrhizal inoculation.

| Carotenoids | Red Fruits | | Green Fruits | |
|---|---|---|---|---|
| | −AM | +AM | −AM | +AM |
| β-carotene | 7.11 ± 1.08 | 9.22 ± 2.14 | 0.64 ± 0.13 | 0.63 ± 0.18 |
| Lycopene | 2.86 ± 0.70 | 4.23 ± 1.96 | n.d. | n.d. |
| Lutein | 1.26 ± 0.40 | 1.23 ± 0.23 | 3.09 ± 0.45 | 3.43 ± 0.47 |
| Zeaxanthin | n.d. | n.d. | 4.37 ± 1.40 | 5.85 ± 1.18 |
| BRIX value | 8.0 ± 0.15 | 8.6 ± 0.09 | 6.8 ± 0.09 | 7.4 ± 0.11 |

Carotenoids were measured using HPLC and are expressed as ng mg$^{-1}$ of dry weight. The fruits from non-mycorrhized (−AM) and mycorrhized (+AM) plants did not significantly differ from one another, and are presented as means ± SD. BRIX values of red fruits with and without mycorrhizae. BRIX was calculated using temperature-adjusted solids percentage. Between the two data sets, Student's $t$-tests found no discernible differences ($p = 0.08$, $n = 8$). Where, n.d. refers to "not detected".

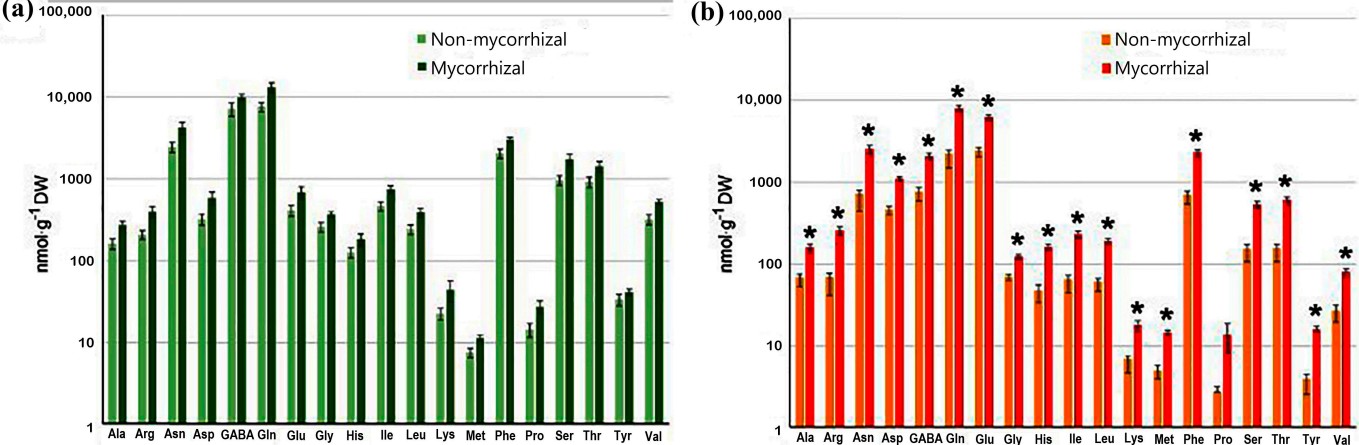

**Figure 2.** Amino acids in fruit tissues with and without mycorrhizae at green (**a**) and red (**b**) growth stages of fruits. The free amino acids present in the methanol-extracted freeze-dried material were determined using HPLC. Note that the amounts of amino acids in green fruits did not differ significantly. All amino acids, except for proline, were elevated in red fruits as a result of mycorrhization of plants. Data are presented as means ± SE ($n = 8$). * demonstrates significant difference between fruits from mycorrhized plants and non-mycorrhized plants at $p < 0.05$.

In the case of 19 different amino acids, no significant difference between mycorrhized plants and non-mycorrhized plants was found in green fruits (Figure 2a). On the other hand, in red fruits, the mycorrhization of plants led to a significant rise in all the amino acid levels except for proline, which remained unchanged (Figure 2b). The levels of glutamine, glutamate, asparagine, and phenylalanine were increased up to four times in the fruit of mycorrhizal plants. Thus, these were the amino acids that showed the greatest increase in concentration.

## 4. Discussion

We demonstrated that a mycorrhizal interaction can be established in an artificial-growth-media-containing cultivation system and can be utilized for large-scale tomato production. The AMF was successfully able to colonize plants leading to increasing levels of amino acids, sugars (increased BRIX values), and carotenoids in the fruit, all of which indicated a higher-quality product. Previous studies demonstrated that the interaction of tomato plants with AMF may result in an increased fruit yield and quality [18,27–29]. Most of them were conducted on plants grown in either a laboratory or an open-field environment with phosphate limitation. The goal of these studies was to examine the possible potential of mycorrhization under greenhouse conditions. In addition, the majority of the substrate for these experiments consisted of quartz sand, which is not an appropriate medium for hydroponics [18,27].

Growers of tomatoes with hydroponic systems have greater control over the growing medium and are better able to preserve optimal nutrient conditions, which leads to increased crop yields and improved fruit quality [3,4]. According to [37–40], a high nutrient supply, such as a high phosphorus level, reduces the interaction. At a high phosphorus availability, an intricate anti-symbiotic syndrome is triggered, which leads to a significant reduction in the amount of root colonization by arbuscular mycorrhizal fungi [41,42]. A slight decline in phosphatic fertilizers (75% of the "normal" phosphorus content) did not affect the yield but allowed *F. mosseae* to colonize the tomato roots, as demonstrated by earlier experiments. The rate of mycorrhiza inoculation that was achieved was lower than in the presence of a more severe phosphorus deficiency; however, the results obtained were adequate for evaluating the influence of mycorrhiza inoculation on fruits. Despite this, our study indicated the influence of arbuscular mycorrhizae in improving the nutritional quality of the tomato fruit without compromising the yield. Additionally, the substrate that was found to be suitable for mycorrhization was a combination of peat and vermiculite. The use of vermiculite in greenhouse tomato production decreases the frequency of mycorrhization of tomato plants even when crushed vermiculite was mixed with clay. This was successful in preventing the mycorrhization of tomato plants. It is currently unknown which components of vermiculite are responsible for producing this effect. There is a possibility that pure coco mats contain antifungal properties, which could have an immediate impact on the AMF. Additionally, this substrate may have high water-percolation properties, which inhibit hyphae from attaching themselves to the root. Mycorrhization was accomplished by utilizing low-phosphate fertilizers in conjunction with the unique substrate mixture; however, it took a few weeks to determine whether or not it had been successful. Many factors can influence this process, particularly, long periods and the comparatively decreased amounts of AMF colonization. Despite the continuous application of fertilizer through drip irrigation, the levels of nitrogen and phosphorus may still be excessively high, which has the potential to disrupt the symbiotic relationship [43]. Further, the yield of the fruit itself is a significant carbon sink due to the limited availability of nutrients for the fungal partner, which would result in a colonization reduction [18].

To improve mycorrhization and provide a solution to these problems, it may be necessary to use alternative substrates and fertilizers, as well as various tomato cultivars and AMF strains that have an increased capacity to colonize tomato plants. This will require further testing of a variety of AMF species. Tomatoes are one of the plant species that shows a positive response to AM in terms of improving fruit characteristics [44,45]. By utilizing the system that was previously described, we were unable to identify any effects of mycorrhization on the production of tomatoes of the cv. Picolino/Maxifour variety. On the other hand, mycorrhizal tomato plants were shown to produce significantly more fruit than non-mycorrhizal tomato plants [18,27,28,46]. It was hypothesized that an interaction with AMF may promote reproductive growth via multifaceted physiological changes. The development of the fruit and flowering processes are accelerated [28], and there are also increases in the overall number of flowers and fruit set [47,48]. It is important to point out, however, that the majority of these studies considerably reduced phosphatic fertilizers, which made it possible for a highly potent and efficient interaction to take place between the plant and the AMF. In contrast to our aims to prevent decreases in the total yield, as a key metric for tomato producers, this would imply that our efforts have been unsuccessful. In addition to this, the tomato cultivar and the species of fungus that are used could both play a crucial role in increasing the yield or not. In this area of study, previous research utilized model tomato cultivars, such as cv. MicroTom [28] and M19 [46], in addition to other species of fungi, such as *F. mosseae* [27,28] and *Glomus fasciculatum* [46]. Because low and late mycorrhization are associated with a slight reduction in phosphorus, there are only minor differences between the transcriptomes of mycorrhizal and non-mycorrhizal plants. This is because non-mycorrhizal plants have higher phosphorus.

The only detectable levels of DEGs were 80 and 51, respectively, in red and green fruits. This is different from what Zouari et al. [48] reported. They found more than

700 DEGs in fruits from mycorrhizal and non-mycorrhizal plants during the stage of ripening transition. Fruits from both mycorrhizal plants and non-mycorrhizal plants were used in this study [48]. In addition, there was no duplication of any of the DEGs that were found here and the DEGs that were found by Zouari et al. [48]. However, among the DEGs identified in the green fruits of the Rio Grande/Nadir varieties, we found genes that may be involved in the processes that regulate fruit ripening (e.g., DEHYDRIN1 and ERF13). In addition, the levels of transcripts for genes that code for proteins that might be associated with fruit nutrition were found to be elevated in green fruits that originated from mycorrhizal plants and displayed differential expression. Among these genes, it was noted that one codes for malic enzyme, another one codes for 2S albumin seed storage protein, one codes for vicilin, and the other one codes for an aminotransferase. The levels of metabolites in red fruits were significantly different, although these variations in green fruits were relatively insignificant. When compared to fruits with non-mycorrhizae-treated plants, fruits produced by mycorrhizae-inoculated plants showed higher free amino acids, lycopene, and BRIX values. The mycorrhiza-mediated improvement in lycopene has also been observed [27,29], even though the molecular mechanisms that are responsible for the increase in carotenoids have not yet been identified [27]. Meanwhile, higher levels of lycopene and β-carotene suggested the positive effect of AM on tomato fruits for human health. Lycopene has a high antioxidant capacity and may have a protective effect on cardiovascular risks, although its efficacy has not been conclusively demonstrated and has exhibited a high degree of variability [49,50]. It is the primary carotenoid found in red tomato fruits [49].

An important finding of this study was an in-depth analysis of free amino acids in the red and green fruit tissues. The results showed that all amino acids were significantly higher accumulated in red fruits derived from mycorrhizal plants, except for proline, when compared to red fruits derived from non-mycorrhizal plants. Glutamic acid, glutamate, phenylalanine, and asparagine were the amino acids that showed significant variation. Only during the ripening stage did inoculated plants indicate higher levels of glutamine and asparagine [28]. These findings suggested that there is a general advantage of using AM to improve the nutritional quality of tomatoes grown in hydroponic greenhouse systems in addition to those grown in traditional soil-based fields. In conclusion, the mycorrhization of commercially grown tomato cultivars in hydroponic systems affects the fruit quality but does not affect the yield, at least in the case of the tomato cultivars and the mycorrhiza strain that were utilized in this study. Similarly, few studies conducted on other crop species, such as strawberry [51], cotton [19], and cucumber [52], found that AM improves the fruit yield and its overall quality. In addition, particular effects on tomato fruit may vary depending on the cultivar of tomato, growth conditions, nutrient applied, and AMF species [3,53–55].

## 5. Conclusions

We conclude that arbuscular mycorrhizal fungi can be used as a promising approach for the production of high-quality tomato fruit on a large-scale. By utilizing a medium that consisted of only trace quantities of vermiculite and phosphate input, our study provides a scientific basis that mycorrhizal colonization can maintain the yield and improve important nutritional quality traits (i.e., sugar, amino acids, lycopene, and carotenoids) of tomato fruit for human consumption. Notably, mycorrhization was accomplished in a commercial greenhouse system that included at least ten times as much nitrogen as phosphorus in its overall composition. Thus, our findings show that inoculation of mycorrhiza can improve the nutritional quality of tomatoes that are grown for commercial purposes in greenhouses.

**Supplementary Materials:** The following supporting information can be downloaded at: https://www.mdpi.com/article/10.3390/horticulturae9040448/s1, Figure S1: Yield of cv. Rio Grande/Nadir tomato plants grown in a commercial greenhouse at Kalar Kahar in 2022; Figure S2: Mineral content of non-mycorrhizal (a) and mycorrhizal (b) green and red fruits; Table S1: List of qRT-PCR primer sequences; Table S2: RNAseq analysis with FPKM values (*n* = 4), log2 fold change (FC), and adjusted

p-value revealed DEGs in red fruits (cv. Nadir) between mycorrhized (+AM) and non-mycorrhized (-AM) plants.

**Author Contributions:** F.U. was involved in the conceptualization, methodology, writing—original draft, and writing—review and editing. H.U. contributed to the methodology. M.I. was involved in the conceptualization and writing—review and editing. S.L.G. contributed to the data analysis. T.K. was involved in the conceptualization and writing—review and editing. Z.L. contributed to conceptualization, supervision, and funding acquisition. All authors have read and agreed to the published version of the manuscript.

**Funding:** This research was funded by the Project of Rural Development of Beijing (project number: BJXCZX20221229) and Beijing Innovation Consortium of Agriculture Research System (project number: BAIC01-2022), and China Scholarship Council (CSC)-Chinese Government Scholarships (CGS).

**Data Availability Statement:** The data presented in this study are available upon request from the corresponding author.

**Acknowledgments:** The authors are grateful for the support and facilitation of China Agricultural University and Agriculture, Live-stalk and Cooperative Department of KPK, Pakistan.

**Conflicts of Interest:** The authors declare that they have no known competing financial interests or personal relationships that could have appeared to influence the work reported in this paper.

## Abbreviations

| | |
|---|---|
| N | Nitrogen |
| P | Phosphorus |
| K | Potassium |
| CV. | Cultivar |
| DW | Dry weight |
| FPKM | Fragments per kilobase of exon per million reads mapped |
| HPLC | High-pressure liquid chromatography |
| Pi | Phosphate |
| qPCR | Quantitative polymerase chain reaction (real-time PCR) |

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
