# Peer review of "Improvement of Nutritional Quality of Tomato Fruit with Funneliformis mosseae Inoculation under Greenhouse Conditions"

_horticulturae, doi:10.3390/horticulturae9040448_

Round 1

Reviewer 1 Report

Comments to the authors are in the attached file.

Author Response

The main purpose of the study was to analyze the effect of mycorrhization on tomato fruit
quality under green house conditions. The subject is original and relevant in the field of
greenhouse tomato production. According to the authors, it is unknown if AMF can improve
the fruit quality of tomato plants that are grown in these conditions.

Abstract it is very long and exceeds 200 words according to journal instructions, so I would
recommend the authors to make it short and more precise. Also the authors should review
the correct way of writing a scientific name, the first letter of genus name is always
capitalized.

---Response: Thanks for the positive comments and valuable suggestions to improve the quality of the submitted manuscript. Following the reviewer’s suggestion and to fulfill the journal requirements, we revised the abstract by providing precise and relevant information. The scientific name is also revised as “Funneliformis mosseae (syn. Glomus mosseae)”

Introduction covered the research point, provides details of research problem, but the
wording is confusing it is suggested to improve it. The authors should review the correct way
of writing a scientific name, it is always written in italics (if typed) and the genus is
“mosseae”.

---Response: To address the reviewer’s concern and make it a quality reading, the introduction part is carefully revised by proving background information about the research problem. All revisions are highlighted in the revised file. The scientific name is also italic, where it is applicable.

Materials and methods chapter is not presented in the order proposed in the instructions for
authors.

---Response: Following the reviewer’s suggestion and to fulfill journal requirements, the M&M section is shifted after the introduction section.

The results of the study were presented confusedly. Some parts of the results section are
written as materials and methods (e.g.lines 113-114). In this section, statements that
correspond to the discussion section are presented (e.g. lines 202-203).

---Response: Thank you for pointing. The mentioned statements have been revised as per suggested.

The discussion should be supported and justified accordingly with some latest references.

---Response: Following the reviewer’s concern, the discussion part has been further improved with the help of the latest references.

Figures and tables needs to be improved, p.e. the use of capital letters in legend titles. it is suggested to comply with the standards of the journal.

---Response:  Thank you for the suggestion. We revised the figures and tables according to journal requirements.

Most of the literature is cited correctly. Considering the quality of the manuscript, it is suggested to rewrite, synthesize it and take a different approach in the presentation of their results and discussion.

---Response:  Thank you for the positive comments. To address the reviewer’s concern and make it a quality reading, we carefully revise the complete manuscript, where it is applicable. Revisions are highlighted in red color.

Author Response

1-Abstract Section, quantify the results, add values to show the difference between the
different treatments.

---Response: Thanks for the valuable suggestions. Following the reviewer’s suggestion, we revised the abstract by providing values of results such as “………… important fruit quality traits such as free amino acids, lycopene (47.9%), β-carotene (29.6%)…………”

2 -In the text, correctly write Glomus mosseae, lines 16 (Abstract), 97 and 325.

---Response: Thanks for highlighting an important point. We revised it as per suggested.

3 -The results obtained clearly show the establishment of the plant/mycorrhizae interaction,
but it was desirable to show this symbiotic association at the end of the experiment by
observing the structures of the endomycorrhizal fungus used: arbuscules, vesicles or others. In
this sense, you also show that the propagules of the initial inoculum are functional and have
completed their cycle with the plant by forming the endomycorrhizal structures in the roots of
the tomato plants.

4-Under other culture conditions, natural conditions, can an endomycorrhizal inoculum based
on a single species induce identical results?

5 - Under natural conditions, will it be necessary to reinforce this inoculum with other inocula or to use a composite inoculum to better ensure mycorrhization of the roots and slow down the effects of competition between the species of inocula and those considered as autochthonous.

---Response: We do understand the reviewer’s concern and valuable suggestions (comments 3-5). The current study was focused on the improvement of fruit nutritional quality traits of tomatoes by Funneliformis mosseae application. We truly appreciate the valuable suggestions of our worthy reviewer and are planned to consider them in our future experiment(s).

Reviewer 3 Report

The authors investigated and discussed “The effects of Glomus mosseae on tomato fruit quality of CV. Rio Grande and Nadir under green house conditions ”. The study is of a great importance and carried out well but there are many changes/clarifications and an extensive revision for language and grammar are needed before publishing. I found the manuscript difficult to follow especially regarding sentence construction. Besides, in some parts, the text lacks flow, which makes difficult to understand the different aspects that the authors report

Title

Please, correct the title to “The effects of Funneliformis mosseae on tomato fruit quality of CV. Rio Grande and Nadir under greenhouse conditions”.

Abstract:

Line 16: Please, add “Funneliformis mosseae (syn. Glomus mosseae)”.

Introduction

This section should be improved and Extensive editing of English language is needed, the authors need to explain how the work is related to other studies found in the published literature. It is required to shorten the introduction; some paragraphs are too long (e.g. lines 31-43). Please, review this section and present the adequate background that is really relevance to the study.

Line 32: “Solanum lycopersicum” in italic

Line 35: (FAO 2018) is a reference should be added as a number.

Line 47-48: I suggest write the statement as “….of fruits to  meet the growing demand for more food with a high content of nutrients ….).

Line 48: “brought about by an expanding human population.” This sentence is not clear. Rephrase it.  

Line 50-51: Please, rewrite this statement “A plant and a fungus that is a member of the phylum Glomeromycotina can form a relationship known as arbuscular mycorrhiza (AM), “then, check the right name of the relationship, is called “arbuscular mycorrhiza (AM)?  

Line 76-77: How Mycorrhization can cut the length of time that tomato plants spend in the vegetative growth stage? Clarify this statement.

Line 86: [Citation needed]?

Line 97+line 100: Write in italic “F. mosseae” instead of “Glomus mossea”

Results

Line 108: This a results section, so why the authors cited two references numbers 13 and 28?

Line 109: “Using the setup described above” which setup? where is mentioned?

Line 138: “Each and every” use only one of the two words not both of them.

Line 151: Delete (+AM).

Line 160-161: This sentence “How sweet a fruit is has a big effect on how it tastes.” is not clear. Please, check the sentence construction

Line 162: “to tell how sweet a fruit is.” The sentence construction should be checked.

Line 194: Delete “s” from amount

Line 198: Figure 5b is missed

Discussions

Line 211: [Citation needed] [Citation needed] ?

Line 220+line 221+line 224+line 227+line 240+line 273+line 275: What is “Pi”?

Line 225:F. mosseae” instead of “Glomus mossea”.

Line 256-257: Start the sentence as “On the other hand, mycorrhizal tomato plants, …”

Line 271: Actually, “Funnelliformis mossea” is synonym of “Glomus mossea” so arrange this sentence.

Line 307: “F. mosseae” instead of “Glomus mossea”

Line 309:  Add “of using….” Instead of “to using…”

Line 316-317: Better to rewrite the sentence as following “In addition, particular effects on tomato fruit may vary depending on the cultivar of tomato, growth conditions, and the AMF species. “

Materials and methods

Line 325:F. mosseae strain BEG12” instead of “Glomus mossea strain BEG12”

Line 380: Rearrange the sentence “Shaking at 70 °C for 30 min extracted 20 mg freeze-dried samples in 400 μL methanol.”

Line 381: “CHCl3”, write the number as superscript “CHCl3

Conclusions

Line 406-408: This sentence needs to be rewritten in a clearer way “We demonstrated that, even in a greenhouse setup, typical of modern industrial indoor tomato production systems, arbuscular mycorrhizal fungi can be used as biostimulants.

Author Response

The authors investigated and discussed “The effects of Glomus mosseae on tomato fruit quality of CV. Rio Grande and Nadir under green house conditions ”. The study is of a great importance and carried out well but there are many changes/clarifications and an extensive revision for language and grammar are needed before publishing. I found the manuscript difficult to follow especially regarding sentence construction. Besides, in some parts, the text lacks flow, which makes difficult to understand the different aspects that the authors report

Title

Please, correct the title to “The effects of Funneliformis mosseae on tomato fruit quality of CV. Rio Grande and Nadir under greenhouse conditions”.

---Response: Thank you for thoroughly reading our manuscript and providing valuable suggestions. Following the reviewer’s suggestion and considering the overall theme of the manuscript, we revised the title as follows: “Improvement of nutritional quality of tomato fruit with Funneliformis mosseae inoculation under greenhouse conditions”

Abstract:

Line 16: Please, add “Funneliformis mosseae (syn. Glomus mosseae)”.

---Response: Thank you for the suggestion. We revised it as per suggested.

Introduction

This section should be improved and Extensive editing of English language is needed, the authors need to explain how the work is related to other studies found in the published literature. It is required to shorten the introduction; some paragraphs are too long (e.g. lines 31-43). Please, review this section and present the adequate background that is really relevance to the study.

---Response: To address the reviewer’s comment and to make it a quality reading, we carefully revised the introduction section by proving relevant and precise background information about the research problem. All revisions are highlighted in the submitted file.

Line 32: “Solanum lycopersicum” in italic

---Response: Revised as per suggested.

Line 35: (FAO 2018) is a reference should be added as a number.

---Response: Number is added as per journal requirement.

Line 47-48: I suggest write the statement as “….of fruits to  meet the growing demand for more food with a high content of nutrients ….).

---Response: Thank you for the suggestion. The given statement has been revised. Now it read as:

“Thus, it is necessary to develop strategies to improve the fertilizer use efficiency of crop plants to meet the growing demand for more food with a high content of nutrients.”

Line 48: “brought about by an expanding human population.” This sentence is not clear. Rephrase it.  

---Response: The mentioned statement has been rephrased. Now complete statement read as:

“Thus, it is necessary to develop strategies to improve the fertilizer use efficiency of crop plants to meet the growing demand for more food with a high content of nutrients.”

Line 50-51: Please, rewrite this statement “A plant and a fungus that is a member of the phylum Glomeromycotina can form a relationship known as arbuscular mycorrhiza (AM), “then, check the right name of the relationship, is called “arbuscular mycorrhiza (AM)?  

---Response: Thank you for your comment, the mentioned statement is rephrased as follows:

“Arbuscular mycorrhiza (AM) is a type of mutualistic symbiosis relationship between a plant and a fungus member of the phylum Glomeromycotina (formerly Glomeromycotina [3]).”

Line 76-77: How Mycorrhization can cut the length of time that tomato plants spend in the vegetative growth stage? Clarify this statement.

---Response: The mentioned statement is clarified as follows:

“Mycorrhization promotes earlier flowering and maturation of the fruit by optimum nutrient supply throughout plant growth [21].”

Line 86: [Citation needed]?

---Response: Relevant citation is provided.

Line 97+line 100: Write in italic “F. mosseae” instead of “Glomus mossea”

---Response: Thank you for the suggestion. Revised throughout the manuscript.

Results

Line 108: This a results section, so why the authors cited two references numbers 13 and 28?

---Response: To address the reviewer’s concern, mentioned two references 13 and 28 have been deleted from the results section.

Line 109: “Using the setup described above” which setup? where is mentioned?

---Response: The confusing statement has been revised as follows:

“To date, there were no significant effects of mycorrhizal inoculation on the greenhouse tomatoes' total yield.”

Line 138: “Each and every” use only one of the two words not both of them.

---Response: Revised as suggested.

Line 151: Delete (+AM).

---Response: (+AM) is deleted in the revised manuscript.

Line 160-161: This sentence “How sweet a fruit is has a big effect on how it tastes.” is not clear. Please, check the sentence construction

---Response: The confusing statement has been deleted in the revised file.

Line 162: “to tell how sweet a fruit is.” The sentence construction should be checked.

---Response: The mentioned statement is revised as follows:

“The BRIX value, which measures the amounts of dissolved solids, mostly sucrose, is a simple way to understand fruit sweetness.”

Line 194: Delete “s” from amount

---Response: “s” is deleted in the revised file.

Line 198: Figure 5b is missed

---Response: Revised as “Figure 3b”.

Discussions

Line 211: [Citation needed] [Citation needed] ?

---Response: Relevant citations are provided in the revised manuscript.

Line 220+line 221+line 224+line 227+line 240+line 273+line 275: What is “Pi”?

---Response: We apologize for the confusion. “Pi” is spelled out as “phosphorus” throughout the manuscript.

Line 225: “F. mosseae” instead of “Glomus mossea”.

---Response: Revised as per suggested.

Line 256-257: Start the sentence as “On the other hand, mycorrhizal tomato plants, …”

---Response: Thank you for the suggestion. A given suggestion is incorporated in the revised manuscript.

Line 271: Actually, “Funnelliformis mossea” is synonym of “Glomus mossea” so arrange this sentence.

---Response: Suggestion has been incorporated in the revised file.

Line 307: “F. mosseae” instead of “Glomus mossea”

---Response: Revised throughout the manuscript.

Line 309:  Add “of using….” Instead of “to using…”

---Response: Revised

Line 316-317: Better to rewrite the sentence as following “In addition, particular effects on tomato fruit may vary depending on the cultivar of tomato, growth conditions, and the AMF species.”

---Response: Suggested revision has been made.

Materials and methods

Line 325: “F. mosseae strain BEG12” instead of “Glomus mossea strain BEG12”

---Response: Revised throughout the manuscript.

Line 380: Rearrange the sentence “Shaking at 70 °C for 30 min extracted 20 mg freeze-dried samples in 400 μL methanol.”

---Response: Rearranged as follows: “Freeze-dried samples were shaken at 70 °C for 30 min in 400 μL methanol.”

Line 381: “CHCl3”, write the number as superscript “CHCl3

---Response: Revised in the submitted file.

Conclusions

Line 406-408: This sentence needs to be rewritten in a clearer way “We demonstrated that, even in a greenhouse setup, typical of modern industrial indoor tomato production systems, arbuscular mycorrhizal fungi can be used as biostimulants.

---Response: We revised as follows: “We concluded that arbuscular mycorrhizal fungi can be used as a promising approach for the production of high-quality tomato fruit on a large-scale”.

Round 2

Reviewer 3 Report

The authors responded the comments and made the necessary modifications but further an extensive revision for language and grammar are needed before publishing. In many parts of the manuscript, the authors added short sentences that can be merged with others (to complete the context) and also used the pronouns “we, they” that should be avoided in the text.

Abstract:

Line 17: Please delete the two commas in …and potassium), fungi, known as arbuscular mycorrhizae (AM), can and keep only that one before “fungi”.

Line 18: Change the word “produce” to “production” in “agricultural produce through higher phosphate uptake”.

Line 28-29: “Taken together, we proposed that mycorrhization can be a promising approach for the production of high-quality tomato fruit for human consumption.” This sentence should be rephrased as following: “Therefore, the current study suggests the mycorrhization as a promising approach for production of high-quality tomato fruit for human consumption”

Introduction

Line 35-36: Rewrite the statement as following “Tomato (Solanum lycopersicum) is one of the most important vegetables for human consumption, especially for the so-called Mediterranean diet, which is low in fat.”

Line 39-40: I suggest write the statement as “Tomatoes are used in a wide range of food and beverage products, not only in their fresh form but also preserves and juice concentrates.”

Line 40-41: I suggest write the statement as “Thought the plants can be commercially grown in greenhouses and open fields, greenhouses are considered ideal for tomato production due to more consistent harvesting throughout the year.

Line 50: “have gain attention” instead of “have drawn attention”

Line 55: Delete “direct”

Line 69-71: I suggest “Thus, AMF are increasingly being incorporated into plant production systems due to a lower use of chemical fertilizers and pesticides, which enhances their contribution to environmentally friendly food production [23].”

Line 75: “during” instead of “throughout

Line 77: Delete “the”

Materials and methods

Line 104: Delete “respectively” and merge the two sentences as: “in 22 × 13 cm pots; filled with a 1:1 mixture of peat and vermiculite.

Line 106-108: Rewrite the sentences as following” One-week old seedlings of cv. Rio Grande and cv. Nadir were inoculated with F. mosseae strain BEG12 [34].

Line 111-112: Which nutrients in “A basic dose of nutrients was applied.” More details should be provided.

Line 111: “applied” is a repeated word already wrote in the previous sentence.

Line 121: “Each truss was matured separately.” is a short sentence. Please try to merge with other one.

Line 159-160: This sentence should be improved “Followed by, they were centrifuged (18,000 g, 10 min, 4 °C) by adding 200 μ L CHCl3 and 400 μ L 160). After “by” add centrifugation and then check the presence of “by” twice in the same sentence.

Line 188-189: Please, rewrite as following “In our study, total plant yield was the same for inoculated and non-inoculated plants (Figure Sup1).

Line 194: Change to “to underline the effect of mycorrhization on the expression pattern of genes

Line 195: Change to “The heatmap showed two distinct categories, for the fruits, represented by the colors green and red respectively.”

Line 196: Change to “This demonstrates that there are substantial differences between the two stages in terms of the expression of genes (Figure 1).

Line 200: Delete “we concluded that

Line 214-216: The word “occurred” noted twice “In addition, variations in gene expression that occurred during the red stage of development might be occurred very late to affect the red fruit’s metabolome. Then, regarding “Therefore, we focused on the DEG levels in green fruits.” this is a short sentence and “we” is also used!.

Line 220: To avoid the repetition of “originated” rewrite as following “… fruits that belong to non-mycorrhizal plants (Table 1).

Line 221: “However, due to the high differences that existed, which were in part caused by the fact that expression was considerably lower in treatment containing no mycorrhiza inoculation.”. The sentence construction should be checked.

Line 234-235: Delete the comma between “solids” and “mostly” in “The BRIX value, which measures the amounts of dissolved solids, mostly sucrose, is a simple way to understand fruit sweetness.

Line 242: “It was” instead of “we” in “we found a significant difference.... “

Line 252: Delete “by” in “… were increased by up to four times in the fruit”

Discussions

Line 266: I suggest rewriting as following “which can be a promising approach frequently utilized by greenhouse tomato growers.

Line 267-268: Please, rewrite as “The AMF was successfully able to colonize plants leading to increasing levels of amino acids, sugars (increased BRIX values), and carotenoids in the fruit, all of which indicated a higher quality product.

Line 269-270: To avoid the repetition of “indicated” rewrite as “Previous studies demonstrated that the interaction of tomato plants with AMF may result in increased fruit yield and quality [18, 27-29].

Line 271: “Most of them” instead of “Most of studies….” to avoid the repetition of “studies”

Line 286-287: Delete this phrase “Despite this, the presence of these nutrient conditions can inhibit mycorrhization”

Line 298: Add “and” between “peat“vermiculite”.

Line 312: The phrase is edited as following “…is a significant carbon sink due to the limit availability of nutrients for the fungal partner, which would result in a colonization reduction [18].

Line 317: The phrase is edited as following “….further testing of a variety of AMF species

Line 328-330: The phrase is edited as following “In contrast to our aims to prevent decreases in total yield, as a key metric for tomato producers, this would imply that our efforts have been unsuccessful”.

Line 330-332: In addition to this, the tomato cultivar and the species of fungus that are used could both play a crucial role in increasing or not the yield.”

Line 334: “Glomus fasciculatum” in italic

Line 339: Edit it as “it was found that more than 700 DEGs… “

Line 343: two “found” in “were found by Zouari et al [48].” and then in Line 344: “we found …”

Line 348-350: Edit it as “Among these genes, it was noted that one codes for malic enzyme, another one codes for 2S albumin seed storage protein, one codes for vicilin, and the other one codes for an aminotransferase

Line 352-353: Edit it as “ ….by mycorrhizae inoculated plants that showed higher free amino acids…”

Line 366: Delete “But” at the beginning of the sentence

Line 371: Edit it as “At least, in case of the tomato cultivar and the mycorrhiza strain that were utilized in this study.”"

Line 373: Delete “the” after “strawberry”

Line 386: Delete “be promising to

Author Response

The authors responded the comments and made the necessary modifications but further an extensive revision for language and grammar are needed before publishing. In many parts of the manuscript, the authors added short sentences that can be merged with others (to complete the context) and also used the pronouns “we, they” that should be avoided in the text.

---Response: Authors are grateful to the worthy reviewer for thoroughly reading our manuscript and helping us to improve the quality of the submitted manuscript.

We have carefully revised the manuscript, and author’s point-by-point response to the reviewer’s comments/suggestions are as follows:

Abstract:

Line 17: Please delete the two commas in …and potassium), fungi, known as arbuscular mycorrhizae (AM), can” and keep only that one before “fungi”.

---Response: Thank you for the correction. Commas are deleted, and now it reads as follows:

Line 17-19: “Apart from important mineral nutrients (i.e., nitrogen, phosphorus, and potassium), fungi known as arbuscular mycorrhizae (AM) can considerably improve the quality of agricultural production through higher phosphate uptake.”

Line 18: Change the word “produce” to “production” in “agricultural produce through higher phosphate uptake”.

---Response: Thank you for the suggestion. In the abstract, “produce” word has been revised to “production”.

Line 28-29: “Taken together, we proposed that mycorrhization can be a promising approach for the production of high-quality tomato fruit for human consumption.” This sentence should be rephrased as following: “Therefore, the current study suggests the mycorrhization as a promising approach for production of high-quality tomato fruit for human consumption”

---Response: Thank you for the suggestion. We have rephrased the sentence in the revised file.

Introduction

Line 35-36: Rewrite the statement as following “Tomato (Solanum lycopersicum) is one of the most important vegetables for human consumption, especially for the so-called Mediterranean diet, which is low in fat.”

---Response: Thank you for your comment. We have rephrased the statement as per suggested.

Line 39-40: I suggest write the statement as “Tomatoes are used in a wide range of food and beverage products, not only in their fresh form but also preserves and juice concentrates.”

---Response: Thank you for your valuable suggestion. Now it reads as follows:

Line 39-41: “Tomatoes are used in a wide range of food and beverage products, not only in their fresh form but also preserves and juice concentrates.”

Line 40-41: I suggest write the statement as “Thought the plants can be commercially grown in greenhouses and open fields, greenhouses are considered ideal for tomato production due to more consistent harvesting throughout the year.”

---Response: To address the reviewer’s concern and to make it a quality reading, we have revised the statement as follows:

Line 41-43: “Though the plants can be commercially grown in greenhouses and open fields, greenhouses are considered ideal for tomato production due to more consistent harvesting throughout the year.”

Line 50: “have gain attention” instead of “have drawn attention”

---Response: “draw” word has been replaced with “gain” in the revised manuscript.

Line 55: Delete “direct”

---Response: Comment has been addressed.

Line 69-71: I suggest “Thus, AMF are increasingly being incorporated into plant production systems due to a lower use of chemical fertilizers and pesticides, which enhances their contribution to environmentally friendly food production [23].”

---Response: Thank you for the suggestion. We have revised the mentioned statement as per suggested.

Line 75: “during” instead of “throughout”

---Response: “throughout” word has been replaced with “during” in the revised manuscript.

Line 77: Delete “the”

---Response: “the” has been deleted in the revised manuscript.

Materials and methods

Line 104: Delete “respectively” and merge the two sentences as: “in 22 × 13 cm pots; filled with a 1:1 mixture of peat and vermiculite.”

---Response: Thank you for your valuable suggestion. Now, the sentence read as follows:

Line 103-105: “The experiment was started from May 2021 to August 2021 and from February 2022 to April 2022 in 22 × 13 cm pots, filled with a 1:1 mixture of peat and vermiculite.”

Line 106-108: Rewrite the sentences as following” One-week old seedlings of cv. Rio Grande and cv. Nadir were inoculated with F. mosseae strain BEG12 [34].

---Response: The mentioned sentence is revised as per suggested.

Line 111-112: Which nutrients in “A basic dose of nutrients was applied.” More details should be provided.

---Response: Following the reviewer’s suggestion and to rule out future confusion, we have provided more detail about the basic dose of nutrients as follows:

Line 111-114: “A basic dose of macro- and micronutrients was applied as follows: 2 mM Ca (NO3)2.4H2O, 2.5 mM CO (NH2)2, 5 mM KNO3, 1 mM KH2PO4, 1 mM MgSO4.7H2O, 23 µM H3BO3, 10 µM MnSO4.H2O, 4.5 µM ZnSO4.7H2O, 0.8 µM CuSO4.5H2O, 0.5 µM H8MoN2O4, and 16 µM EDTA-FeNa.3H2O.”

Line 111: “applied” is a repeated word already wrote in the previous sentence.

---Response: Thank you for your concern. Now it reads as:

Line 114-115: “The nutrient solution was provided during the first few months of plant growth, with as few as three irrigation cycles and as many as thirty after that.”

Line 121: “Each truss was matured separately.” is a short sentence. Please try to merge with other one.

---Response: We have revised it as follows:

Line 122-123: “Fruits were allowed to ripen by selectively pruning leaves at a rate of no more than three leaves per week, and each truss was matured separately.”

Line 159-160: This sentence should be improved “Followed by, they were centrifuged (18,000 g, 10 min, 4 °C) by adding 200 μ L CHCland 400 μ L 160). After “by” add centrifugation and then check the presence of “by” twice in the same sentence.

---Response: Thank you for your concern. Now it reads as:

Line 162-163: “Subsequently, they were centrifuged (18,000 g, 10 min, 4 °C) after adding 200 µL CHCl3 and 400 µL distilled water (18.2 cm),……..”

Line 188-189: Please, rewrite as following “In our study, total plant yield was the same for inoculated and non-inoculated plants (Figure Sup1).

---Response: Thank you for your valuable suggestion. We have revised it as per suggested.

Line 194: Change to “to underline the effect of mycorrhization on the expression pattern of genes”

---Response: We have revised following the reviewer’s suggestion.

Line 195: Change to “The heatmap showed two distinct categories, for the fruits, represented by the colors green and red respectively.”

---Response: We have revised it, and now it reads as:

Line 198-199: “The heatmap showed two distinct categories, for the fruits, represented by the colors green and red respectively.”

Line 196: Change to “This demonstrates that there are substantial differences between the two stages in terms of the expression of genes (Figure 1).

---Response: We have revised the statement as per suggested.

Line 200: Delete “we concluded that”

---Response: “we concluded that” has been deleted in the submitted file.

Line 214-216: The word “occurred” noted twice “In addition, variations in gene expression that occurred during the red stage of development might be occurred very late to affect the red fruit’s metabolome. Then, regarding “Therefore, we focused on the DEG levels in green fruits.” this is a short sentence and “we” is also used!.

---Response: To address the reviewer’s concern, we rewrite the sentence as follows:

Line 216-218: “In addition, variations in gene expression during the red stage of development might be occurred very late to affect the red fruit’s metabolome. Therefore, DEG levels were focused in green fruits.”

Line 220: To avoid the repetition of “originated” rewrite as following “… fruits that belong to non-mycorrhizal plants (Table 1).

---Response: The comment has been addressed in the revised file.

Line 221: “However, due to the high differences that existed, which were in part caused by the fact that expression was considerably lower in treatment containing no mycorrhiza inoculation.”. The sentence construction should be checked.

---Response: To address the reviewer’s concern, we revised the sentence as follows:

Line 222-223: “However, overall gene expression was considerably lower in non-mycorrhizal plants.”

Line 234-235: Delete the comma between “solids” and “mostly” in “The BRIX value, which measures the amounts of dissolved solids, mostly sucrose, is a simple way to understand fruit sweetness.”

---Response: Following the reviewer’s suggestion, the comma between “solids” and “mostly” has been deleted.

Line 242: “It was” instead of “we” in “we found a significant difference.... “

---Response: The comment has been addressed in the revised file.

Line 252: Delete “by” in “… were increased by up to four times in the fruit”

---Response: “by” has been addressed in the submitted file.

Discussions

Line 266: I suggest rewriting as following “which can be a promising approach frequently utilized by greenhouse tomato growers.”

---Response: To address the reviewer’s concern, we revised the sentence as follows:

Line 266-267: “………….., and can be utilized for large-scale tomato production.”

Line 267-268: Please, rewrite as “The AMF was successfully able to colonize plants leading to increasing levels of amino acids, sugars (increased BRIX values), and carotenoids in the fruit, all of which indicated a higher quality product.”

---Response: The sentence has been revised as per suggested (Line 264-266).

Line 269-270: To avoid the repetition of “indicated” rewrite as “Previous studies demonstrated that the interaction of tomato plants with AMF may result in increased fruit yield and quality [18, 27-29].”

---Response: We revised it in the submitted file (Line 267-269).

Line 271: “Most of them” instead of “Most of studies….” to avoid the repetition of “studies”

---Response: We replaced “Most of studies….” with “Most of them” it in the revised manuscript.

Line 286-287: Delete this phrase “Despite this, the presence of these nutrient conditions can inhibit mycorrhization”

---Response: The mentioned statement has been deleted from the submitted file.

Line 298: Add “and” between “peat” “vermiculite”.

---Response: Thank you for the correction. “and” has been added between “peat” “vermiculite”.

Line 312: The phrase is edited as following “…is a significant carbon sink due to the limit availability of nutrients for the fungal partner, which would result in a colonization reduction [18].

---Response: Thank you for the suggestion. The mentioned statement has been edited as per suggested (Line 310-312).

Line 317: The phrase is edited as following “….further testing of a variety of AMF species

---Response: Thank you for the suggestion. The mentioned sentence has been revised in the submitted file.

Line 328-330: The phrase is edited as following “In contrast to our aims to prevent decreases in total yield, as a key metric for tomato producers, this would imply that our efforts have been unsuccessful”.

---Response: We revised it in the submitted file (Line 328-330).

Line 330-332: “In addition to this, the tomato cultivar and the species of fungus that are used could both play a crucial role in increasing or not the yield.”

---Response: Thank you for suggestion. We revised it.

Line 334: “Glomus fasciculatum” in italic

---Response: “Glomus fasciculatum” is revised to italic.

Line 339: Edit it as “it was found that more than 700 DEGs… “

---Response: Comment has been addressed in the revised manuscript.

Line 343: two “found” in “were found by Zouari et al [48].” and then in Line 344: “we found …”

---Response: Thank you for pointing that out. Now it reads as follows:

Line 338-340: “This is different from what Zouari et al. [48] reported. It was found that more than 700 DEGs in fruits from mycorrhizal and non-mycorrhizal plants during the stage of ripening transition.”

Line 348-350: Edit it as “Among these genes, it was noted that one codes for malic enzyme, another one codes for 2S albumin seed storage protein, one codes for vicilin, and the other one codes for an aminotransferase”

---Response: Thank you for your valuable suggestion. We revised the statement in the submitted file.

Line 352-353: Edit it as “ ….by mycorrhizae inoculated plants that showed higher free amino acids…”

---Response: We revised as per suggested.

Line 366: Delete “But” at the beginning of the sentence

---Response: “But” has been deleted in the revised file.

Line 371: Edit it as “At least, in case of the tomato cultivar and the mycorrhiza strain that were utilized in this study.”"

---Response: Thank you for the correction. We edited the sentence as per suggested.

Line 373: Delete “the” after “strawberry”

---Response: “the” has been deleted after “strawberry”

Line 386: Delete “be promising to”

---Response: Thank you for the correction. Deleted in the submitted file.
